# Geometric Organization of Cognitive States in Transformer Embedding Spaces

## Abstract

Recent work has shown that transformer-based language models learn rich geometric structure in their embedding spaces. In this work, we investigate whether sentence embeddings exhibit structured geometric organization aligned with human-interpretable cognitive or psychological attributes. We construct a dataset of 480 natural-language sentences annotated with both continuous energy scores (ranging from $-5$ to $5$) and discrete tier labels spanning seven ordered cognitive annotation tiers, intended to capture a graded progression from highly constricted or reactive expressions toward more coherent and integrative cognitive states. Using fixed sentence embeddings from multiple transformer models, we evaluate the recoverability of these annotations via linear and shallow nonlinear probes. Across models, both continuous energy scores and tier labels are reliably decodable, with linear probes already capturing substantial structure. To assess statistical significance, we conduct nonparametric permutation tests that randomize labels, showing that probe performance exceeds chance under both regression and classification null hypotheses. Qualitative analyses using UMAP visualizations and tier-level confusion matrices further reveal a coherent low-to-high gradient and predominantly local (adjacent-tier) confusions. Together, these results indicate that transformer embedding spaces exhibit statistically significant geometric organization aligned with the annotated cognitive structure.

## 1 Introduction

Modern transformer-based language models (Vaswani et al., 2017) encode text as points in high-dimensional representation spaces that capture rich semantic and contextual structure (Ethayarajh, 2019; Reimers & Gurevych, 2019). These embedding spaces have been studied in applications such as semantic similarity, sentence-level understanding, and affective analysis (Demszky et al., 2020; Mohammad, 2018). Recent work in representation analysis shows that abstract semantic properties can form geometric structure within language model representations. Studies have identified approximately linear organization of spatial, temporal, and conceptual attributes in LLM representations (Gurnee & Tegmark, 2024; Park et al., 2023). Empirical findings also suggest that specific variables such as sentiment polarity or truthfulness may align with separable directions or low-dimensional regions in model representation spaces (Hollinsworth et al., 2024; Marks & Tegmark, 2024). While several prior studies examine individual semantic attributes, it remains an open question whether graded cognitive states correspond to a broader geometric organization across embedding space.

In this work, we investigate whether transformer sentence embeddings exhibit geometric organization aligned with graded cognitive annotations. We evaluate separability and local geometric adjacency among annotated attributes, treating the annotations as probes into representation geometry, and examine whether cognitively relevant signals align with approximately linear or low-dimensional structure in embedding space. Rather than solely isolating individual semantic directions, this study focuses on the broader geometric organization of cognitively annotated language within sentence embedding space.

To study this question, we construct a dataset of 480 natural-language sentences annotated along a seven-tier taxonomy of cognitive states, ranging from low-awareness, high-distress expressions to high-coherence,

integrative states. The annotation scheme draws conceptual inspiration from psychological and contemplative literature (Varela et al., 1992; Jung, 1964; Hawkins, 2002; Laozi, 1891), but is used here solely as an operational labeling framework for empirical analysis rather than as a comprehensive theoretical framework.

We analyze multiple widely used sentence embedding models, including BGE, MPNet, and MiniLM, using a combination of visualization, probing (Alain & Bengio, 2016; Conneau et al., 2018; Hewitt & Liang, 2019; Belinkov & Glass, 2019), directional ablation, and statistical validation techniques. Specifically, we employ UMAP to examine the global geometry of the embeddings, linear and shallow nonlinear probes to quantify the recoverability of the annotated cognitive attributes, and nonparametric permutation tests (Good, 2013) to assess whether observed patterns exceed chance under label randomization.

Our results provide evidence that transformer sentence embeddings exhibit a statistically significant geometric organization correlated with the proposed cognitive attribute annotations and continuous energy scores. These findings suggest that beyond surface-level sentiment, embedding spaces may reflect structured representation patterns aligned with human-interpretable cognitive and psychological states.

**Contributions.** This work makes the following contributions:

(1) We built a graded cognitive annotation scheme for natural-language sentences, consisting of seven ordered tiers designed to probe geometric organization in sentence embedding spaces. (2) We assemble and integrate a set of established geometric analysis techniques into a unified evaluation pipeline, providing a concrete test for investigating linearity and structured organization. (3) We present a geometric perspective on cognitively annotated language, showing that embedding representations exhibit consistent organization across models and analysis methods, suggesting that representation geometry can serve as a measurable substrate for studying structured cognitive attributes in language.

## 2  Dataset

To examine whether transformer embedding spaces exhibit structured geometric organization aligned with graded cognitive annotations, we construct a manually annotated dataset of natural-language sentences spanning a wide range of affective and cognitive expressions.

### 2.1  Cognitive Tier Taxonomy

Inspired by work in cognitive science, psychology, consciousness studies, and contemplative traditions, (Varela et al., 1992; Jung, 1964; Hawkins, 2002; Laozi, 1891) we define a seven-tier taxonomy representing qualitatively distinct modes of cognitive experience. The tiers are ordered from highly contracted, self-destructive states to expansive, integrative states:

- **Shadow (Unconscious / Collapse):** Ignorance, self-blame, despair, apathy, psychological collapse.

- **Striving (Scarcity and Attachment):** Fear, craving, insecurity, anxiety, survival-oriented thinking.

- **Conflict (Ego and Opposition):** Anger, hostility, dominance, control, power struggles.

- **Activation (Energy Mobilization):** Courage, resolve, neutrality, acceptance, behavioral readiness.

- **Growth (Inner Reorganization):** Openness, forgiveness, transformation, contribution, learning.

- **Clarity (Cognitive Integration):** Reasoning, abstraction, understanding, coherence, meaning.

- **Unity (Non-Dual Integration):** Compassion, joy, peace, surrender, wholeness.

Each tier represents a mode of organization of experience rather than merely emotional valence.

## 2.2 Sentence Construction and Annotation

The dataset consists of 480 short natural-language sentences distributed across the seven tiers. Sentences were constructed to reflect:

- predominantly first-person, introspective, or experiential language, with higher-tier sentences adopting more contemplative or aphoristic phrasing that reflects increasingly abstract or integrative cognitive framing;

- limited reliance on explicit emotion words (e.g., "happy", "sad", "angry") as primary labeling cues.

For example, the sentence "I keep working harder so no one can say I failed" reflects a striving-oriented cognitive stance without relying on explicit emotion terminology. Similarly, "I'm learning to step back before reacting" illustrates regulation and transition as a shift in cognitive framing rather than a direct expression of sentiment. In higher tiers, more contemplative phrasing may naturally appear (e.g., references to stillness or connection), such as "Stillness is not absence, it's fullness without motion." Explicit emotion or sentiment terms were not strictly prohibited, but were used when necessary to preserve natural semantic coherence.

The annotations were performed manually by the author to ensure internal consistency across tiers. The objective was semantic representativeness rather than exhaustive coverage. Each sentence is assigned to exactly one tier, and no sentence appears in multiple tiers. Table 1 shows example sentences representing each cognitive tier. Additional examples are provided in Appendix A

Table 1: Representative example sentences from each cognitive tier with illustrative energy scores.

| TIER | EXAMPLE SENTENCE | Score |
|------|------------------|-------|
| Shadow | "I feel like everything I do just makes things worse, and I don't see a way out." | $-4.5$ |
| Striving | "I keep worrying that I'm not doing enough, and they'd leave me." | $-2.9$ |
| Conflict | "Why would I listen to people not at my level? Nobody knows better than me." | $-1.7$ |
| Activation | "I can accept what is happening and pull myself back to center." | $0.0$ |
| Growth | "I'm learning from what happened and trying to respond differently this time." | $1.8$ |
| Clarity | "Looking at the situation objectively helps me understand why it unfolded this way." | $3.0$ |
| Unity | "I feel a quiet sense of connection and compassion, even in difficulty." | $4.2$ |

## 2.3 Continuous Energy Scores

In addition to discrete tier labels, each sentence is annotated with a continuous energy score ranging from $-5$ to $+5$. These scores provide a coarse ordinal signal reflecting the relative contraction or expansion of the cognitive state expressed by the sentence. The energy score is intended as a human-interpretable scalar capturing how language varies in perceived activation, vitality, or depletion. Energy scores were assigned manually by the author to reflect the relative position of each sentence along the proposed low-to-high cognitive spectrum. Scores were chosen to be internally consistent within and across tiers, and are allowed to overlap across adjacent tiers, particularly near tier boundaries:

- Lower values (approximately $-5$) correspond to highly contracted, self-destructive states.

- Higher values (approximately $+5$) correspond to expansive, integrative states.

- Intermediate values correspond to transitional or neutral states, with scores near 0 reflecting activation or readiness, marking a shift from contracted toward more expansive modes along this scale.

## 2.4 Dataset Intent

The continuous energy scores and tier labels are not treated as precise measurements or interval-scaled ground truth. Instead, they serve as a coarse ordinal signal used to examine whether sentences expressing similar cognitive attributes occupy nearby regions in embedding space and exhibit consistent relative ordering.

The dataset is intended for empirical analysis of representation geometry in the present work. While future research may explore broader interpretive or assessment-oriented applications, the current study does not propose or validate any clinical, diagnostic, or psychological evaluation framework.

# 3 Methods

## 3.1 Sentence Embeddings

We study whether pretrained language models encode structured geometric organization aligned with cognitively annotated attributes in their embedding spaces. Given a dataset of natural-language sentences annotated with both discrete tier labels (Section 2.1) and continuous energy scores (Section 2.3), we first map each sentence into a fixed-dimensional embedding space using frozen pretrained encoders.

We evaluate three widely used sentence embedding models commonly used in sentence representation learning:

- `BAAI/bge-large-en-v1.5` (Xiao et al., 2023)

- `sentence-transformers/all-mpnet-base-v2` (Song et al., 2020)

- `sentence-transformers/all-MiniLM-L6-v2` (Wang et al., 2020)

All models are used in inference-only mode without any fine-tuning. Each sentence is encoded into a single vector representation using the model's default pooling strategy, and all subsequent analyses operate on these fixed embedding representations.

## 3.2 Probing Analysis

To evaluate whether cognitive structure is encoded in transformer embedding spaces, we employ a set of probing models that predict annotated attributes from fixed sentence embeddings (Alain & Bengio, 2016; Conneau et al., 2018; Hewitt & Liang, 2019; Belinkov & Glass, 2019)obtained from the models described in Section 3.1. Probing is used as a diagnostic tool to assess which information is recoverable from the representations.

### 3.2.1 Regression Probes for Continuous Energy Scores

Each sentence is annotated with a continuous energy score in the range $[-5, 5]$, reflecting its position along a low-to-high cognitive spectrum (Section 2.3). We examine whether this scalar signal is encoded in embedding geometry using two regression probes.

**Linear probe (Ridge regression).** Ridge regression provides a conservative test of whether energy scores are linearly decodable from embeddings, serving as a lower bound on representational structure while controlling for overfitting through $\ell_2$ regularization.

**Nonlinear probe (MLP regressor).** To assess the presence of additional nonlinear structure, we train a shallow multilayer perceptron with two hidden layers. This probe captures modest nonlinear interactions while remaining limited in capacity, avoiding the expressivity of deep task-optimized models.

Models are trained using an 80/20 train–test split and evaluated using the coefficient of determination ($R^2$) and mean squared error (MSE).

### 3.2.2 Classification Probes for Cognitive Tiers

In addition to continuous scores, each sentence is labeled into one of seven ordered cognitive tiers: Shadow, Striving, Conflict, Activation, Growth, Clarity, and Unity (Section 2.1). We train a multiclass logistic regression classifier and a shallow multilayer perceptron (MLP) to predict tier labels from sentence embeddings.

Logistic regression is deliberately chosen as a low-capacity linear classifier to ensure that classification performance reflects intrinsic separability of tier structure in the embedding space rather than probe expressivity. Performance is reported using accuracy and weighted F1-score to account for class imbalance. We additionally evaluate a shallow nonlinear probe using a small MLP classifier. This auxiliary analysis tests whether allowing limited nonlinear decision boundaries substantially changes the observed results.

### 3.2.3 Confusion Matrix Analysis

To further analyze classification behavior, we inspect confusion matrices of the tier classifier. Confusion matrices are visualized for a representative train–test split (random seed = 0).

We examine whether misclassifications predominantly occur between adjacent tiers (e.g., Growth $\leftrightarrow$ Clarity, Activation $\leftrightarrow$ Growth), rather than between distant tiers (e.g., Striving $\leftrightarrow$ Clarity). Such locality-sensitive errors would indicate that embedding geometry preserves ordinal structure aligned with the graded nature of the tier annotations.

### 3.2.4 Stability Across Splits

For each embedding model, probing results are averaged over 30 random train–test splits with an 80/20 split ratio. Each split uses a distinct random seed, controlling both data partitioning and model initialization (for nonlinear probes). Reported metrics correspond to mean performance across splits, reducing variance due to sampling effects and providing a more robust estimate of probe behavior.

### 3.3 UMAP Visualization

To provide a qualitative view of the geometric organization of cognitive annotations in embedding space, we apply Uniform Manifold Approximation and Projection (UMAP) (McInnes et al., 2018) to each embedding set. Points are colored by the continuous energy score in the range $[-5, 5]$ using a continuous colormap. We report 3D UMAP visualizations in the main paper and include 2D UMAP plots as supplementary material in Appendix C.

### 3.4 Permutation Tests

High probe performance alone does not guarantee that embeddings encode target attributes in a meaningful way; strong results may arise from incidental correlations or dataset artifacts. To assess whether the observed probing performance reflects a genuine alignment between embedding geometry and annotated attributes, we conduct nonparametric permutation tests under a label-randomization null (Good, 2013). We use a fixed random number generator (RNG) seed in permutation tests to ensure reproducibility and fair comparison between the observed statistic and the null distribution.

### 3.4.1 Null Hypotheses and Test Statistics

We consider two complementary null hypotheses:

- **Energy score null (energy score regression).** Continuous energy scores are independent of sentence embeddings.

- **Tier null (tier classification).** Discrete tier labels are independent of sentence embeddings.

Under each null, labels are randomly permuted while embeddings $X$ are held fixed. For each permuted dataset, we re-run the probing protocol from Section 3.2 using 30 repeated 80/20 train–test splits with fixed split seeds.

We use the following test statistics:

- Regression: mean Ridge $R^2$ across splits, $\overline{R^2}$.

- Classification: mean weighted F1-score across splits, $\overline{\text{F1}_{\text{w}}}$.

Permutation tests follow a Monte Carlo procedure described in Appendix D.

### 3.5  Directional Ablation of Energy-Related Structure

To examine whether energy-related information is organized along a specific geometric direction in embedding space, we perform a direction-based ablation inspired by prior work on linear representation features and directional analyses in language model embeddings (Hollinsworth et al., 2024; Gurnee & Tegmark, 2024).

First, a linear regression probe (Ridge regression) is trained to predict the continuous energy score from sentence embeddings. Let $X \in \mathbb{R}^{N \times d}$ denote the embedding matrix ($N$ sentences, $d$-dimensional embeddings) and $y \in \mathbb{R}^N$ the corresponding energy scores. The fitted weight vector $w \in \mathbb{R}^d$ is interpreted as a candidate energy-related direction. We normalize this vector to obtain a unit direction $\hat{w} = w/\|w\|_2$.

Directional ablation is then performed by removing the projection of each embedding onto $\hat{w}$:

$$x' = x - (x \cdot \hat{w})\hat{w}. \tag{1}$$

This operation eliminates only the component aligned with the learned direction while preserving all orthogonal information. The resulting embeddings are treated as fixed inputs and evaluated using the same probing procedures described in Sections 3.2 and subsequent analyses.

To quantify the geometric importance of the removed direction beyond task-based probe performance, we measure the fraction of total embedding variance captured along $\hat{w}$. Let $X_c$ denote mean-centered embeddings. The variance along the direction is computed as

$$\text{Var}_{\hat{w}} = \text{Var}(X_c \hat{w}), \tag{2}$$

and normalized by the trace of the covariance matrix,

$$\text{Ratio} = \frac{\text{Var}_{\hat{w}}}{\text{trace}(\text{Cov}(X_c))}. \tag{3}$$

This quantity reflects how much of the overall geometric structure of the embedding space is aligned with the ablated direction.

As controls, we also perform ablations using randomly sampled directions and directions learned from permuted energy labels, testing whether the observed effects arise from removing arbitrary linear projections rather than from the annotation-aligned direction itself.

### 3.6  TF-IDF Baseline Representation

We include TF-IDF as a non-contextual lexical baseline evaluated using the same probing protocol as contextual embeddings, reporting regression, classification, confusion matrix, and UMAP analyses. This baseline helps distinguish patterns captured by contextual semantic representations from those that can be explained by word statistics alone.

## 4 Results

### 4.1 Decodability of Continuous Energy Scores

We first evaluate whether the continuous energy scores assigned to sentences are recoverable from fixed transformer embeddings. This analysis tests whether embedding geometry preserves graded structure aligned with a low-to-high cognitive spectrum.

Across all evaluated embedding models, energy scores are strongly decodable. As shown in Table 2, regression performance is well above chance, capturing a substantial proportion of the annotated ordinal signal on held-out data. For the BAAI/bge-large-en-v1.5 embeddings, the mean coefficient of determination exceeds 0.80, indicating that a large fraction of the annotated energy signal is recoverable from the representation space. Comparable but slightly lower performance is observed for `all-mpnet-base-v2` and `all-MiniLM-L6-v2`.

Table 2: Energy regression probe performance averaged over 30 train–test splits.

| Model | Ridge $R^2$ ↑ | Ridge MSE ↓ | MLP $R^2$ ↑ | MLP MSE ↓ |
|---|---|---|---|---|
| BAAI/bge-large-en-v1.5 | 0.808 | 1.824 | 0.830 | 1.605 |
| all-mpnet-base-v2 | 0.750 | 2.373 | 0.769 | 2.182 |
| all-MiniLM-L6-v2 | 0.671 | 3.118 | 0.698 | 2.859 |

Decodability improves modestly when moving from linear to shallow nonlinear regression. Across all models, nonlinear probes achieve slightly higher $R^2$ and lower mean squared error than their linear counterparts.

However, this improvement should be interpreted cautiously. While it may indicate that some energy-score-related variation is not perfectly captured by a single linear direction, it may also reflect the coarse and manually assigned nature of the annotation scheme. Because energy scores are intended as approximate ordinal signals rather than precise interval-scaled measurements, shallow nonlinear models may partially compensate for annotation noise or boundary ambiguity rather than revealing intrinsic nonlinear structure in the embedding space.

We also observe a clear ordering across embedding models. Larger and more expressive models yield stronger decodability, with BGE outperforming MPNet, and MPNet outperforming MiniLM. This monotonic trend holds for both linear and nonlinear probes, indicating that representational capacity influences how clearly graded energy information is preserved.

Overall, these results provide strong evidence that continuous energy scores are consistently recoverable from transformer embedding spaces, suggesting that graded energy-related structure is reflected in their geometry.

### 4.2 Decodability of Cognitive Tiers

We next examine whether the discrete cognitive tiers assigned to sentences are recoverable from transformer embedding representations. Unlike the continuous energy scores in Section 4.1, tier labels represent coarser categorical stages along the same underlying spectrum.

Across all embedding models, tier labels are substantially decodable, with classification performance well above chance. As summarized in Table 3, weighted F1-scores range from approximately 0.70 to 0.77 across models, indicating that the embedding space preserves meaningful separation among the seven tiers.

Consistent with the regression results, performance varies systematically across models. BAAI/bge-large-en-v1.5 achieves the strongest tier decodability, followed by `all-mpnet-base-v2` and `all-MiniLM-L6-v2`. This ordering mirrors the trend observed for continuous energy prediction, suggesting that both continuous and categorical annotations align with shared representational structure.

To better understand the nature of classification errors, we inspect confusion matrices for a representative train–test split. Figure 1 shows the confusion matrix for tier classification using BAAI/bge-large-en-v1.5 embeddings. Misclassifications are concentrated between adjacent tiers (e.g., Activation ↔ Growth, Growth

Table 3: Tier classification probe performance averaged over 30 train–test splits. Shallow nonlinear probes were implemented using small multilayer perceptrons. Several lightweight hidden-layer configurations were explored as a robustness check; results shown correspond to a representative configuration (128,64), which yielded performance comparable to linear probes across models.

| | Logistic Regression | | MLP (128,64) | |
| Model | Accuracy ↑ | Weighted F1 ↑ | Accuracy ↑ | Weighted F1 ↑ |
| --- | --- | --- | --- | --- |
| BAAI/bge-large-en-v1.5 | 0.779 | 0.766 | 0.771 | 0.763 |
| all-mpnet-base-v2 | 0.774 | 0.764 | 0.766 | 0.756 |
| all-MiniLM-L6-v2 | 0.717 | 0.703 | 0.692 | 0.675 |

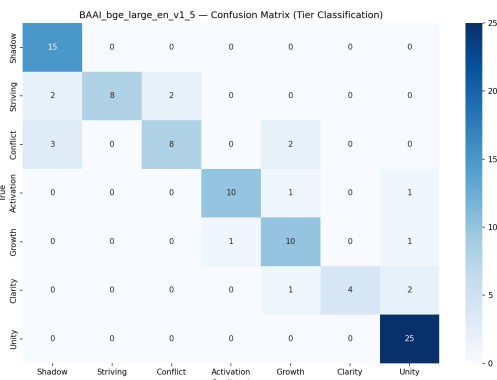

Figure 1: Confusion matrix for linear tier classification using BAAI-bge-large-en-v1.5 embeddings.

↔ Clarity), while confusions between distant tiers (e.g., Striving ↔ Clarity) are rare. Similar patterns are observed for other embedding models (Appendix B).

Taken together, these results indicate that cognitive tiers are not only decodable, but are organized in an ordered fashion within the embedding space. The concentration of errors between neighboring tiers suggests that embeddings encode a graded structure consistent with the proposed tiers.

## 4.3   Qualitative Structure in Embedding Space via UMAP

To complement quantitative probing results, we examine the geometric organization of sentences in embedding space using UMAP visualizations (McInnes et al., 2018) colored by continuous energy scores (Figure 2).

Across all three embedding models, UMAP reveals a clear low-to-high energy gradient rather than random mixing. Sentences annotated with lower energy scores (e.g., Shadow and Striving) tend to occupy contiguous regions, while higher-energy sentences (e.g., Clarity and Unity) occupy distinct regions of the embedding space.

Model-dependent differences are apparent. BAAI/bge-large-en-v1.5 exhibits the most coherent structure, with a smooth, approximately monotonic transition from low-energy to high-energy regions. `all-mpnet-base-v2` shows a similar global gradient but with increased overlap between adjacent energy levels. `all-MiniLM-L6-v2` displays greater dispersion and mixing, consistent with its weaker regression and classification performance.

Importantly, misalignments observed in the confusion matrices—primarily between adjacent tiers—are reflected in UMAP by local overlaps rather than long-range mixing. Distant tiers (e.g., Shadow vs. Unity) rarely occupy the same regions, suggesting that embedding geometry preserves a coarse order even when fine-grained boundaries are ambiguous.

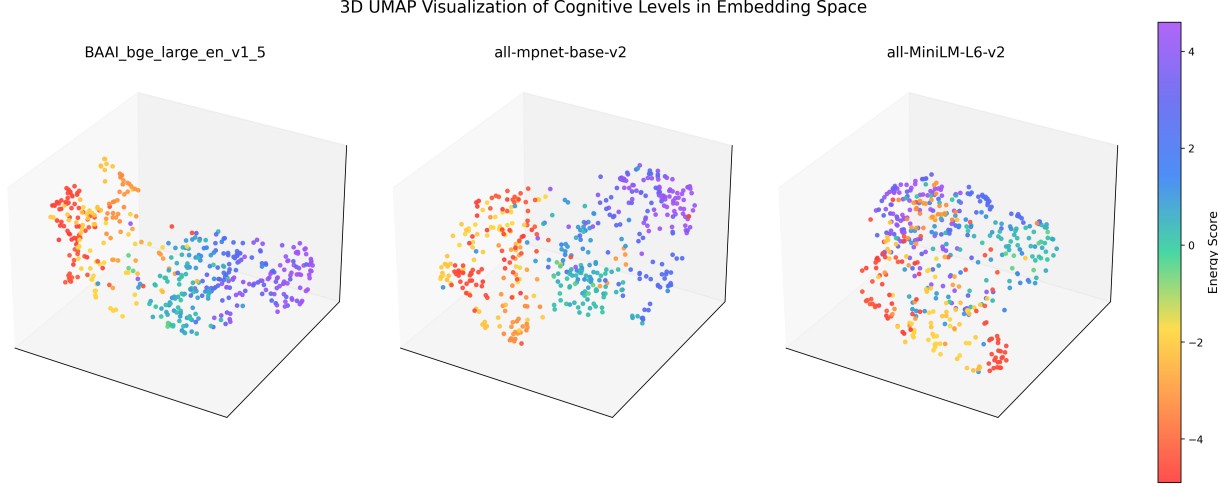

Figure 2: 3D UMAP visualization of sentence embeddings colored by energy scores.

## 4.4 Statistical Significance via Permutation Tests

To assess whether the observed probe performance reflects a genuine alignment between embedding geometry and the annotated cognitive attributes—rather than spurious correlations—we evaluate statistical significance using nonparametric permutation tests (Section 3.4).

Table 4: Permutation test results across embedding models ($N_{\mathrm{perm}} = 200$). Observed probe performance consistently exceeds shuffled-label baselines.

| Model | Ridge $R^2$ | $p$-value | Tier F1 | $p$-value |
|---|---|---|---|---|
| BAAI/bge-large-en-v1.5 | 0.808 | $< 0.005$ | 0.776 | $< 0.005$ |
| all-mpnet-base-v2 | 0.750 | $< 0.005$ | 0.761 | $< 0.005$ |
| all-MiniLM-L6-v2 | 0.671 | $< 0.005$ | 0.706 | $< 0.005$ |

Across embedding models, observed probe performance averaged across 30 random 80/20 train–test splits lies far outside the permutation null distributions. Under the score permutation null, the empirical distribution of mean Ridge $R^2$ values remains well below the observed statistics, yielding a one-sided permutation $p$-value of $p_{\mathrm{score}} < 0.005$ with $N = 200$ permutations. A similar pattern is observed under the tier permutation null, where weighted F1 scores consistently exceed shuffled-label baselines ($p_{\mathrm{tier}} < 0.005$).

As shown in Figure 3, empirical null distributions remain concentrated near chance performance, while observed probe statistics lie in the extreme upper tail.

While all-mpnet-base-v2 and all-MiniLM-L6-v2 yield lower absolute scores than BAAI/bge-large-en-v1.5, observed performance for all models remains well separated from permutation null distributions. In both score and tier settings, fewer than 1 in 200 shuffled label assignments achieve comparable results ($p < 0.005$). Histogram visualizations for these models are provided in Appendix E

Together, these findings indicate that embedding representations encode information aligned with both continuous energy scores and discrete cognitive tiers. The consistency of permutation significance across embedding models suggests that the observed separability reflects robust geometric organization rather than model-specific effects.

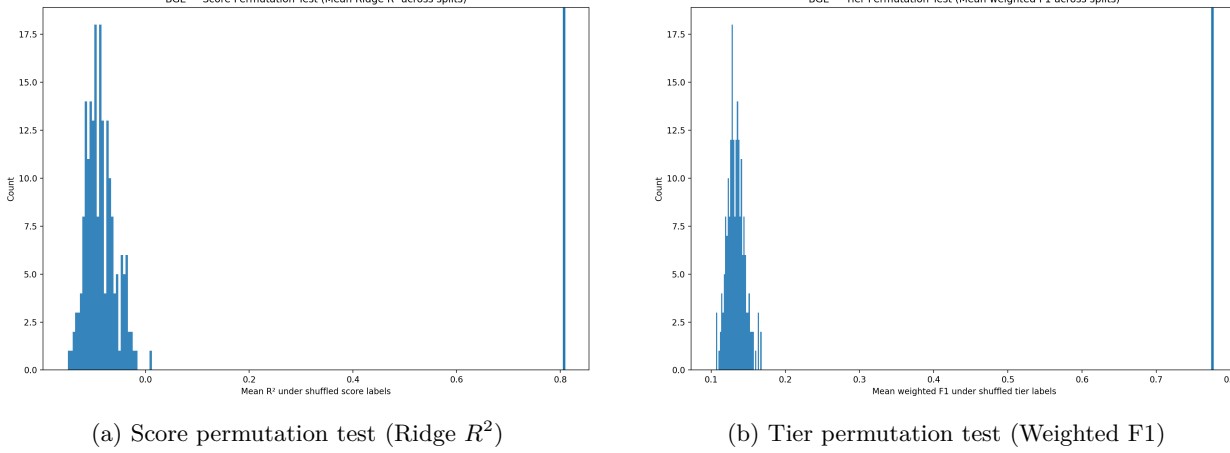

(a) Score permutation test (Ridge $R^2$)    (b) Tier permutation test (Weighted F1)

Figure 3: Permutation test results for score regression and tier classification. Vertical lines indicate observed probe performance.

## 4.5 Directional Ablation

We next evaluate how removing the learned energy-aligned direction affects downstream probing performance. This analysis tests whether energy-related information is concentrated along a specific geometric component in embedding space.

Table 5 compares energy regression performance before and after directional ablation. Removing the learned direction substantially reduces regression performance across all models. For example, Ridge $R^2$ drops from 0.808 to 0.452 for `bge-large-en-v1.5`, indicating that a significant portion of the energy-related signal is concentrated along a specific geometric component. Nonlinear probes exhibit a similar decline.

Table 5: Energy regression probe performance under directional ablation. Normal denotes original embeddings, Energy-removed denotes projection removal along the learned energy direction, and Perm-label control denotes removal of a direction learned from permuted labels. Results are averaged over 30 train–test splits.

| Model | Condition | Ridge $R^2$ ↑ | Ridge MSE ↓ | MLP $R^2$ ↑ | MLP MSE ↓ |
|---|---|---|---|---|---|
| BAAI/bge-large-en-v1.5 | Normal | 0.808 | 1.824 | 0.830 | 1.605 |
| | Energy-removed | 0.452 | 5.207 | 0.498 | 4.773 |
| | Perm-label ctrl | 0.807 | 1.828 | 0.829 | 1.617 |
| all-mpnet-base-v2 | Normal | 0.750 | 2.373 | 0.769 | 2.182 |
| | Energy-removed | 0.344 | 6.219 | 0.446 | 5.246 |
| | Perm-label ctrl | 0.747 | 2.395 | 0.768 | 2.196 |
| all-MiniLM-L6-v2 | Normal | 0.671 | 3.118 | 0.698 | 2.859 |
| | Energy-removed | 0.220 | 7.407 | 0.349 | 6.186 |
| | Perm-label ctrl | 0.670 | 3.132 | 0.697 | 2.869 |

Additional analyses are reported in Appendix F. Tier classification performance shows a modest decline following directional ablation, indicating that the removed component contributes to, but does not solely determine, the geometric organization associated with the tier annotations. This contrast with the larger drop observed in energy regression suggests that tier separability is supported by multiple distributed geometric features rather than a single dominant direction. Variance analysis shows that the learned direction accounts for a measurable share of embedding variance, consistent with a geometrically prominent feature rather than a negligible component. In contrast, ablations along randomly sampled directions or directions learned from permuted labels produce minimal changes in probe performance. Taken together, these findings indicate that the observed performance drop is selectively associated with the annotation-aligned direction rather than a generic consequence of dimensionality reduction.

### 4.6 TF-IDF Baseline

Table 6: Comparison between contextual embedding models and a TF–IDF lexical baseline. Results are averaged over 30 train–test splits.

| Model | Ridge $R^2$ ↑ | Ridge MSE ↓ | Tier Accuracy ↑ | Weighted F1 ↑ |
|---|---|---|---|---|
| BAAI/bge-large-en-v1.5 | 0.808 | 1.824 | 0.779 | 0.766 |
| all-mpnet-base-v2 | 0.750 | 2.373 | 0.774 | 0.764 |
| all-MiniLM-L6-v2 | 0.671 | 3.118 | 0.717 | 0.703 |
| TF-IDF (1–2gram) | 0.444 | 5.296 | 0.473 | 0.421 |

As shown in Table 6, the TF–IDF baseline performs substantially worse than contextual embedding models across both regression and classification tasks, suggesting that the observed structure cannot be accounted for by lexical statistics alone. Additional TF–IDF analyses and visualizations are provided in Appendix G.

## 5 Discussion

### 5.1 What Is Encoded in the Embedding Geometry?

Across multiple embedding models, both continuous energy scores and discrete cognitive tiers are reliably decodable from fixed sentence embeddings, suggesting a graded geometric organization aligned with the annotated attributes. For continuous energy prediction, linear probes recover a substantial portion of the signal, while shallow nonlinear probes provide moderate additional improvements, consistent with energy-related structure that is largely aligned with global geometric directions but not purely linear. In contrast, for tier classification, linear probes perform comparably to or slightly better than nonlinear probes, suggesting that categorical boundaries are well approximated by linear separations in embedding space.

The observed organization is unlikely to be explained solely by lexical statistics. A TF–IDF baseline shows substantially lower performance than contextual embedding models across both regression and classification tasks, indicating that the measured structure is not recoverable from lexical frequency features alone. Permutation tests further demonstrate that probe performance falls well outside the label-randomization null distribution, supporting the interpretation that the alignment reflects consistent geometric structure rather than chance associations.

Directional ablation further supports this interpretation. Removing a learned energy-aligned direction substantially reduces probe performance despite accounting for only a small fraction of total embedding variance. This disproportionate impact suggests that predictive signal is concentrated along specific geometric directions rather than being uniformly distributed across embedding dimensions.

### 5.2 Global Ordering and Local Ambiguity

Both quantitative and qualitative analyses suggest that embedding space preserves a coarse global ordering of the annotated attributes while allowing for local ambiguity. Confusion-matrix analyses show that misclassifications primarily occur between neighboring tiers (e.g., Activation vs. Growth), whereas confusions between distant categories (e.g., Shadow vs. Unity) are relatively rare. UMAP visualizations exhibit a similar pattern, with local overlap among nearby energy levels and increasing separation across more distant regions.

Such behavior is consistent with a graded representation containing fuzzy boundaries between adjacent human-defined categories. Given the subjective and context-dependent nature of cognitive or psychological language, partial overlap between neighboring annotations is expected. This pattern suggests that embedding geometry captures a continuous organization in which discrete labels approximate regions along a shared representational spectrum.

### 5.3 Model Differences and Representational Capacity

Comparisons across embedding models reveal systematic differences in how strongly this structure is expressed. Larger and more expressive models (e.g., BAAI/bge-large-en-v1.5) exhibit higher probe performance and clearer geometric organization in low-dimensional projections. Smaller models (e.g., MiniLM) show greater dispersion and overlap, consistent with reduced representational capacity.

These model-dependent differences suggest that the emergence of graded structure in embedding space is influenced by representational capacity and training characteristics. At the same time, the presence of above-chance decodability across all evaluated models indicates that this organization reflects a broader property of transformer-based sentence representations, with model scale influencing the clarity and robustness of the encoded geometry rather than its existence.

## 6 Conclusion and Future Work

This work examines how human-defined cognitive annotations relate to representation geometry in transformer sentence embeddings. Results show that both continuous energy scores and discrete tier labels correspond to approximately linear organization within embedding space. Instead of being evenly distributed across dimensions, cognitively annotated signals appear concentrated along specific geometric directions that remain consistent across multiple embedding models. Linear probes capture a large portion of this structure, while shallow nonlinear probes provide modest improvements, suggesting that the dominant organization is accessible through relatively simple geometric features.

From a geometric perspective, one consistent pattern across analyses is that sentences associated with similar cognitive states tend to occupy nearby regions in embedding space. UMAP projections, probe performance, and confusion matrix structures collectively suggest that local neighborhoods preserve similarity in annotated attributes, while transitions between adjacent tiers occur gradually rather than through abrupt boundaries.

These findings suggest several directions for future research. The localization of cognitively annotated language within specific regions of embedding space may support representation-level safety analysis and interpretability, where embedding geometry serves as a diagnostic signal for analyzing model behavior. For example, localized regions could help study shifts toward higher-risk or coercive linguistic patterns without relying solely on keyword-based filtering. More broadly, the observed geometric organization motivates exploration of representation-space monitoring or steering approaches, in which generation trajectories are analyzed relative to geometric regions associated with different attributes. Future work may further evaluate robustness across datasets, languages, and annotation paradigms, explore alternative cognitive taxonomies, and investigate whether similar graded organization emerges within intermediate transformer layers or during generation dynamics.

## 7 Limitations

The cognitive tier labels and continuous energy scores are manually annotated by a single annotator and applied to a dataset of limited size and scope consisting of short English sentences. While this design ensures internal consistency, it may introduce subjectivity and limit generalization. Future work may assess robustness using multi-annotator agreement, alternative annotation schemes, larger or more naturalistic corpora, and additional languages.

### Broader Impact Statement

Our findings suggest that embedding spaces may contain localized regions associated with psychologically sensitive or unstable states. While such structure could support beneficial applications such as interpretability or safety analysis, misuse may arise if representation geometry is deliberately leveraged to steer model behavior toward unstable regions.

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

# A    Additional Example Sentences per Tier

This appendix provides additional illustrative example sentences for each tier to support qualitative interpretability. These examples are intended to help readers develop an intuitive sense of the cognitive–affective states indexed by each tier, independent of access to the full annotated dataset.

**Tier 1: Highly Contracted / Self-Destructive, Apathy, Despair**

- "The feeling was like standing in a tunnel with no light at either end."
- "I can't stop blaming myself for what happened."
- "I don't care anymore - well, nobody cares about me anyways."

**Tier 2: Contracted / Striving, Fear, Manipulation**

- "I keep myself busy so I don't feel the heaviness underneath."
- "If I can make them doubt their memory, I stay safe."
- "I sense danger behind their words, attention, and actions, so I play it safe by only saying what they want to hear."

**Tier 3: Defensive / Conflict, Ego, Anger**

- "I hate seeing his name pop up; it instantly ruins my mood."
- "I feel attacked easily and respond by pushing back."
- "Ugh, I don't mix with people beneath me."

**Tier 4: Activated / Transitional, Accepting, Courage**

- "Whatever comes, I'll face it one step at a time."
- "I'm not waiting for permission anymore; I start to trust my voice."
- "I accept what has already happened and focus on what I can do next."

**Tier 5: Constructive / Growth-Oriented, Relaxed, Healing**

- "I keep my word, even when it's hard."
- "I want to give because I'm becoming someone who can give, and what I share can actually make a difference."
- "No matter how busy I am, I take breaks to relax my eyes and body."

**Tier 6: Integrated / Harmonious, Clarity, Reasoning**

- "Comprehension feels like light turning on inside my mind."

- "I can see the pattern beneath the surface."

- "Let's examine the possible causes of the issue piece by piece."

**Tier 7: Expansive / Unifying, Grace, Wisdom**

- "Love doesn't ask why; it simply shines."

- "Even difficulty feels part of a larger whole."

- "When the self dissolves, nothing is abandoned and nothing is gained."

**Note** These examples are provided for reference only. All analyses reported in the main paper are based exclusively on the annotated dataset described in the Dataset Section.

## B    Confusion Matrices for Tier Classification Across Three Embedding Models.

This appendix presents confusion matrices for tier classification across all evaluated embedding models (Figure 4), illustrating that misclassifications occur predominantly between adjacent tiers rather than distant ones. The stronger models exhibit sharper diagonal structure.

## C    2D UMAP visualization of continuous energy scores

This appendix provides 2D UMAP visualizations of sentence embeddings colored by continuous energy scores (Figure 5), offering a qualitative view of the graded geometric structure discussed in the main text.

## D    Monte Carlo Permutation Procedure

Let $T_{\text{obs}}$ denote the observed test statistic. We approximate the null distribution via Monte Carlo permutation by repeating the following procedure $N = 200$ times:

1. Randomly permute target labels $y' \leftarrow \pi(y)$ using a fixed random number generator seed.

2. Apply the same probing protocol as in Section 3.2 using the same split seeds.

3. Compute the mean test statistic $T_i$ across splits.

A one-sided permutation $p$-value is computed using the smoothed estimator (Good, 2013):

$$p = \frac{1 + \sum_{i=1}^{N} \mathbb{I}[T_i \geq T_{\text{obs}}]}{N + 1}.$$

## E    Additional Permutation Test Results

This appendix presents permutation test histogram for all-mpnet-base-v2 and all-MiniLM-L6-v2 models.

(1) Model: all-mpnet-base-v2 (Figure 6)

(2) Model: all-MiniLM-L6-v2 (Figure 7)

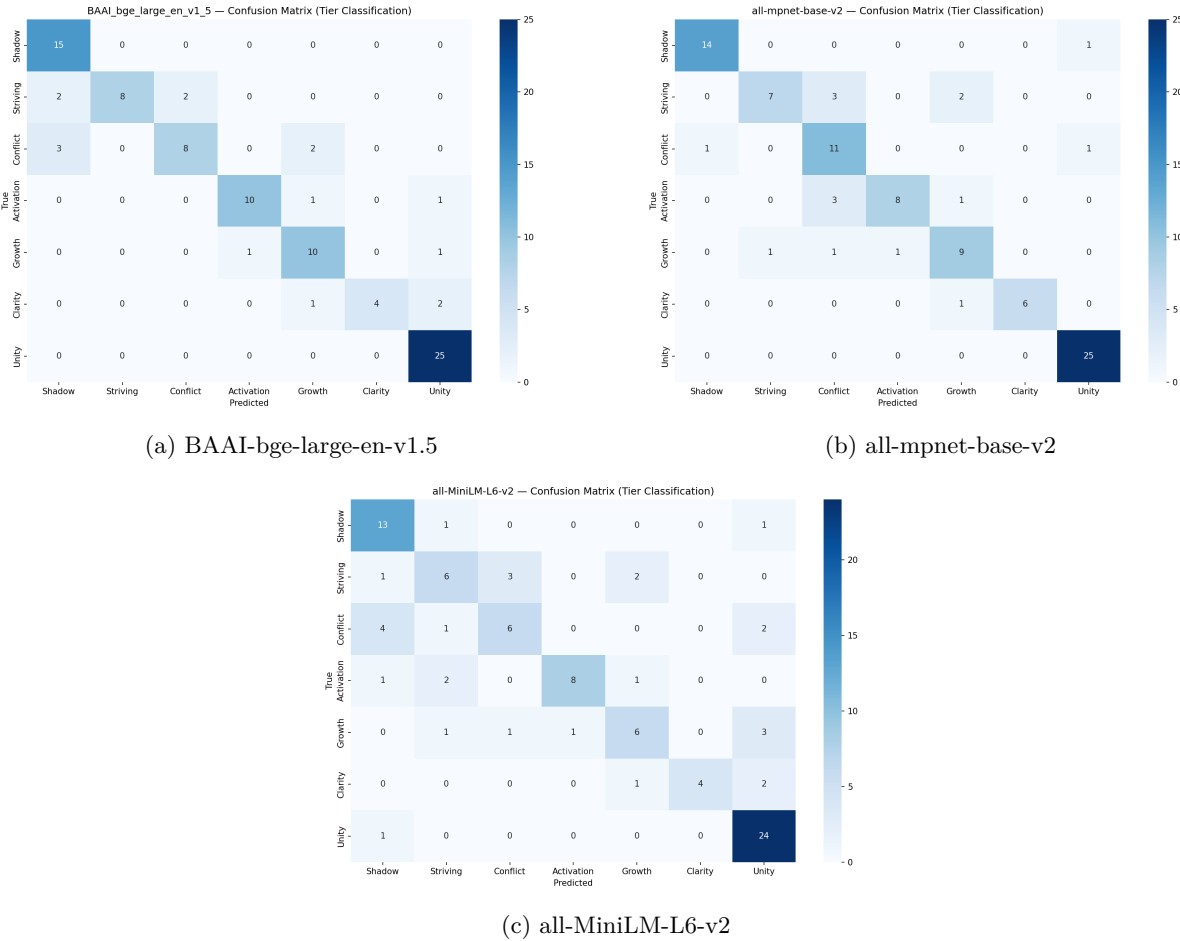

(a) BAAI-bge-large-en-v1.5      (b) all-mpnet-base-v2

(c) all-MiniLM-L6-v2

Figure 4: Confusion matrices for linear tier classification across three embedding models.

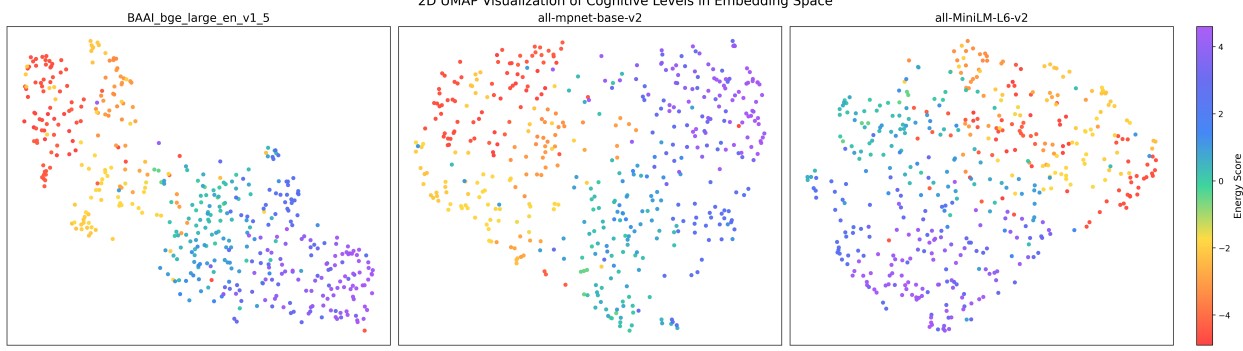

Figure 5: 2D UMAP visualization of sentence embeddings colored by continuous energy scores.

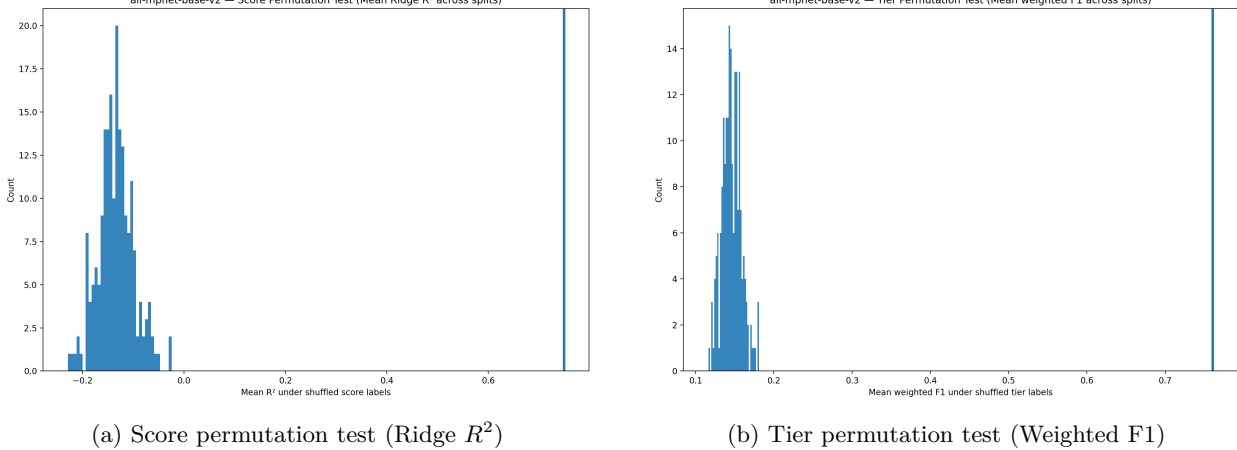

(a) Score permutation test (Ridge $R^2$)    (b) Tier permutation test (Weighted F1)

Figure 6: Permutation test results for score regression and tier classification (all-mpnet-base-v2). Vertical lines indicate observed probe performance.

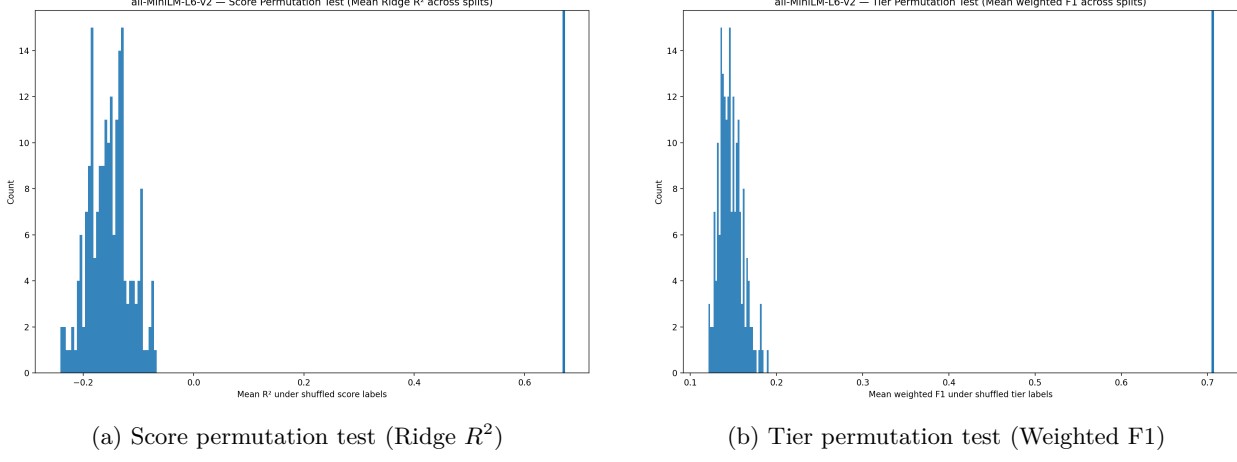

(a) Score permutation test (Ridge $R^2$)    (b) Tier permutation test (Weighted F1)

Figure 7: Permutation test results for score regression and tier classification (all-MiniLM-L6-v2). Vertical lines indicate observed probe performance.

## F    Additional Directional Ablation Analyses

This appendix presents additional analyses supporting the directional ablation results reported in Section 4.5. Specifically, we examine (i) the impact of ablation on tier classification performance, (ii) the geometric prominence of the learned direction, (iii) control ablations using randomly sampled directions, and (iv) control ablations using permuted-label directions.

### F.1    Effect on Tier Classification

A comparable but smaller decrease is observed for tier classification (Table 7), where weighted F1-scores decline by approximately 4–9 points depending on the model. This pattern indicates that the ablated direction contributes not only to continuous score prediction but also to the broader organization of cognitive annotations within embedding space. Notably, the reduction is less pronounced than in the regression task, suggesting that classification decisions rely on additional geometric features beyond a single dominant direction.

Table 7: Tier classification performance before and after directional ablation, averaged over 30 train–test splits.

|  | Logistic Regression | | MLP (128,64) | |
| Model | F1 (Orig.) ↑ | F1 (Ablated) ↑ | F1 (Orig.) ↑ | F1 (Ablated) ↑ |
| --- | --- | --- | --- | --- |
| BAAI/bge-large-en-v1.5 | 0.766 | 0.717 | 0.763 | 0.724 |
| all-mpnet-base-v2 | 0.764 | 0.703 | 0.756 | 0.697 |
| all-MiniLM-L6-v2 | 0.703 | 0.614 | 0.675 | 0.609 |

### F.2    Geometric Prominence of the Learned Direction

To quantify how strongly the learned direction contributes to the overall embedding geometry, we measure the variance captured along $\hat{w}$ relative to the total embedding variance, computed as the trace of the covariance matrix. Formally, we evaluate

$$\text{Ratio} = \frac{\text{Var}(X_c\hat{w})}{\text{trace}(\text{Cov}(X_c))},$$

where $X_c$ denotes mean-centered embeddings.

As shown in Table 8, the learned direction accounts for approximately 2%–4% of total variance across models. Although modest in absolute magnitude, this variance share is substantially larger than that of random directions sampled in the same embedding space, where typical ratios remain below 0.3%. The learned energy direction lies at the extreme tail of the random-direction distribution (100th percentile across models), indicating that the ablated component corresponds to a statistically prominent geometric feature rather than an arbitrary projection.

Table 8: Variance share of the learned energy direction relative to the total embedding variance (trace of the covariance matrix).

| Model | Var(direction) | Trace(Cov) | Ratio (%) |
| --- | --- | --- | --- |
| BAAI/bge-large-en-v1.5 | 0.0224 | 0.5067 | 4.43 |
| all-mpnet-base-v2 | 0.0211 | 0.8197 | 2.57 |
| all-MiniLM-L6-v2 | 0.0173 | 0.8256 | 2.10 |

### F.3    Control Ablation by Removing Random Directions

For each embedding model, we sample 500 random unit directions and compute the variance-share statistic $\text{Var}(X_c u)/\text{trace}(\text{Cov}(X_c))$ for each direction $u$. Across all three models, the energy-aligned direction exhibits

a substantially larger variance share than randomly sampled directions. Concretely, the mean random-direction variance share is $9.7 \times 10^{-4}$ (bge-large), $1.29 \times 10^{-3}$ (mpnet), and $2.61 \times 10^{-3}$ (MiniLM), whereas the learned energy direction attains $4.43 \times 10^{-2}$, $2.57 \times 10^{-2}$, and $2.10 \times 10^{-2}$ respectively. In all cases, the energy-direction variance share exceeds every one of the 500 sampled random directions (empirical percentile = 100% in our samples), suggesting that the ablated component corresponds to a geometrically prominent direction rather than an arbitrary projection.

### F.4 Control Ablation Using Permuted Labels

To verify that the observed effects are not simply caused by removing any linear direction, we repeat the ablation procedure using a direction learned from permuted energy labels. Under this control condition, probe performance remains largely unchanged compared to the original (non-ablated) embeddings (Table 5), indicating that the regression collapse observed in Section 4.5 is more closely associated with the annotation-aligned direction rather than a generic dimensionality reduction effect.

### F.5 Summary

Taken together, these analyses suggest that cognitively annotated attributes are not uniformly distributed across embedding space but instead exhibit partial alignment with a localized geometric direction.

## G Additional Analyses for the TF–IDF Baseline

We performed additional analyses using a TF–IDF baseline. These analyses mirror the evaluation protocol used for contextual embeddings but are reported here in detail for completeness.

### G.1 Experimental Setup

Sentences are represented using a TF–IDF vectorizer with unigram and bigram features, ngram_range = $(1, 2)$, min_df = 2, max_df = 0.95, and sublinear term-frequency scaling. All evaluation procedures follow the same protocol as described in Section 3.2, including 30 repeated 80/20 train–test splits.

### G.2 Energy Regression and Tier Classification

Table 9 reports detailed regression and classification metrics for the TF–IDF baseline. Compared to contextual embedding models, TF–IDF representations yield substantially lower performance across all tasks, indicating that lexical frequency patterns alone do not capture the structured organization observed in embedding space.

Table 9: TF–IDF baseline performance averaged over 30 train–test splits.

| Metric | Linear Probe | MLP Probe |
|---|---|---|
| Energy regression ($R^2$) ↑ | 0.444 | 0.457 |
| Energy regression (MSE) ↓ | 5.296 | 5.165 |
| Tier classification (Accuracy) ↑ | 0.473 | 0.487 |
| Tier classification (Weighted F1) ↑ | 0.421 | 0.446 |

### G.3 Confusion Matrix Analysis

Figure 8 shows the tier classification confusion matrix for the TF–IDF baseline (seed 0). Compared to contextual embeddings, the TF–IDF representation exhibits weaker separation between adjacent tiers and more diffuse misclassification patterns, suggesting reduced structural coherence.

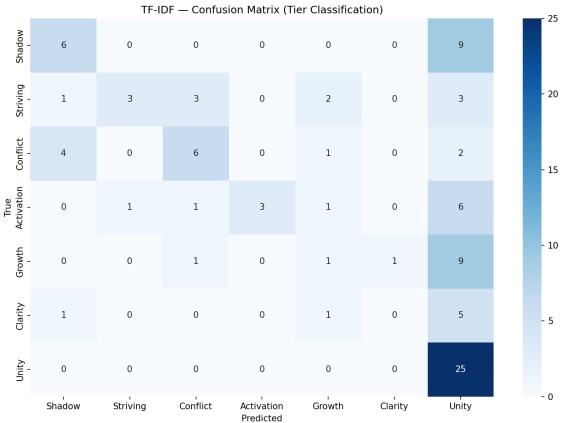

Figure 8: TF–IDF linear tier classification confusion matrix.

## G.4 UMAP Visualization

To provide a qualitative comparison with contextual embeddings, we visualize TF–IDF representations using 2D and 3D UMAP projections (Figures 9a and 9b). Compared to contextual embeddings, TF–IDF representations exhibit weaker low-to-high energy gradients and less coherent geometric organization across tiers, indicating that the structure observed in contextual models is not explained by lexical frequency features alone.

Notably, TF–IDF projections still display a faint energy gradient, suggesting that lexical usage partially correlates with the annotated scores. This provides empirical support for the internal linguistic consistency of the annotation scheme, while the substantially stronger organization observed in contextual embeddings indicates that additional structure is captured beyond lexical statistics.

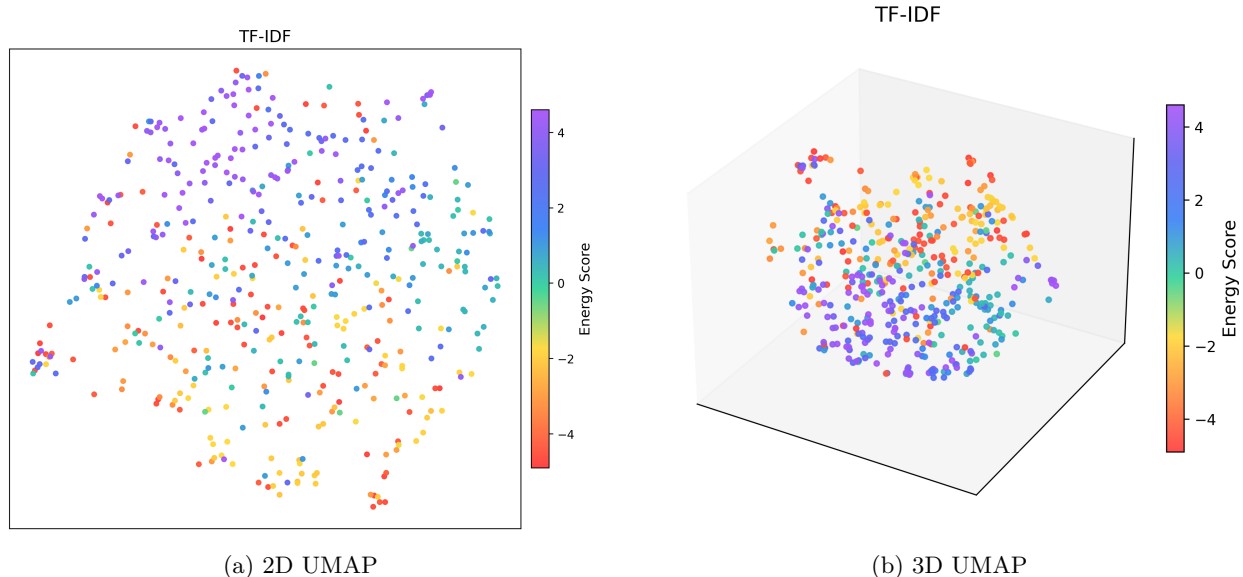

(a) 2D UMAP        (b) 3D UMAP

Figure 9: UMAP visualizations of TF–IDF representations colored by energy score.

