# OpenReview forum: "Geometric Organization of Cognitive States in Transformer Embedding Spaces"
_TMLR — Rejected by TMLR_

### Review · Reviewer_zQPT · 2026-01-31

**Summary Of Contributions:**

This work studies the organizational structure of sentence embeddings in Transformer-based models. Specifically, whether said embeddings exhibit hierarchical and graded organization of psychologically relevant states. To this end the authors introduce an annotated dataset of sentences which they have designed to encompass a set of pre-defined tiers of mental states and the corresponding values within each of said tiers. Using different visualization techniques they then explore how these embeddings are organized.

**Audience:**

Yes

**Audience Explanation:**

I imagine people interested on the linearity of embedding spaces in any model (not just Transformers) would find this useful as a test case for what methods to apply to validate their claims.

**Broader Impact Concerns:**

No concerns.

**Claims And Evidence:**

No

**Claims Explanation:**

I am not sure about what the claim regarding hierarchy. First I thought that each tier would have their own internal scaling of the energy function,  but based on the description of the analysis, it seems that they are just discretization of the energy spectrum. I think it's interesting the everything lies on a mostly linear subspace, but this is different from being hierarchical.

**Requested Changes:**

I would say the claims of hierarchy are misleading. I would suggest either dropping it or finding a test which does show it. For a two-level hierarchy this could be something like the graphs you get when Simpson's paradox occurs: there are two tiers of variables, say male and female and within each tier there are different values (like height and weight). The slope of the variables may not vary within each of the two classes but the bias does. Would such a hierarchical prediction model fair better than the linear regression one on the whole embedding space?

---

> ### Author Response · Authors · 2026-02-26
> **Response to Reviewer zQPT**
>
> We thank the reviewer for raising important concerns regarding the term "hierarchy." We agree that the current wording may suggest a stronger claim than what is empirically supported; our intention was not to assert a strict hierarchical structure. In Revision 1, we have removed the hierarchical language and instead adopted more neutral descriptions, such as "geometric organization" or "spectrum."
>
> Again, we appreciate the reviewer for sharing their suggestions and expertise with us! Please let us know if there is anything further we should clarify.

---

### Review · Reviewer_fvGm · 2026-02-06

**Summary Of Contributions:**

This paper investigates whether sentence embeddings encode a graded, hierarchical structure. To study this question the authors create a list of attributes then assign them ranks in a hierarchy according to psychological and philosophical ideas. Then, they provide evidence showing that the attributes adjacent in the hierarchy are represented in adjacent parts of models' representation spaces.

**Additional Comments:**

"The localization of lower-energy or high-risk psychological language within specific regions of embedding space may support applications in safety analysis, interpretability, and the differentiation of coercive versus consent-aligned language."

I don't understand this - what applications could be supported?

"More broadly, the observed geometry points toward the possibility of non-manipulative generation steering, in which model outputs are guided toward regions of embedding space associated with higher coherence or alignment-related attributes without reliance on explicit rule-based filtering"

I don't understand this - in what sense could steering be called "non-manipulative"?

**Audience:**

Yes

**Audience Explanation:**

Yes, people new to the linear representation hypothesis may be interested to know how it applies in this particular case.

**Claims And Evidence:**

No

**Claims Explanation:**

No. They state that their evidence indicates that embeddings encode hierarchical structure. However, even concepts that the authors would likely not assign to a hierarchy are represented in this way, as adjacent to each other. The canonical example here is the male -> female spectrum in a language embedding model, where male and female embeddings are adjacent. The "hierarchy" in the proposed taxonomy seems to be a purely human invention. Therefore this paper seems conceptually flawed.

If the properties in the taxonomy were assigned adjacencies rather than ranks the evidence would fit more neatly. If "hierarchy" were replaced with "spectrum" the evidence may also fit more neatly.

Example quote: "Both quantitative and qualitative analyses point to a hierarchical organization of the annotated tiers within embedding space"

**Requested Changes:**

Remove the claims around hierarchy and replace them with something like spectrum, or adjacency. This is critical to securing recommendation for acceptance, unless my own claims are somehow conceptually flawed.

Contextualize the work using at least one other more recent language model representation paper. I found https://arxiv.org/pdf/2310.02207 which focuses on space and time representations but there are many good ones.

Not critical to my recommendation:

- Causal ablations are pretty standard in probing papers
- Citing some of the many probing papers from '23 onwards would help contextualize this work more accurately
- "meaningfulness" is not well-defined and a strong definition would probably entail causal ablations
- In "This structure cannot be explained by surface lexical statistics alone" "surface" is a load-bearing term that is not defined. "indicating that the observed organization reflects higher-order contextual representations rather than word frequency or n-gram co-occurrence patterns" seems inaccurate to me because n-grams can be of arbitrary order, with 2-grams and higher being contextual. A skip-10-gram is higher order than a 1-gram.

---

> ### Author Response · Authors · 2026-02-26
> **Response to Reviewer fvGm**
>
> We thank the reviewer for the thoughtful feedback and questions! We truly enjoyed the constructive suggestions and have incorporated them into revision 1.
>
> **Major changes include:**
>
> - Removed hierarchical terminology and replaced it with more neutral descriptions such as geometric organization or spectrum. We agree that our evidence supports graded organization or spectrum rather than a strict hierarchy.
> - Contextualized the study with additional recent work on representation geometry in language models. ( in the first paragraph of section 1 Introduction )
> - Added directional-ablation experiments together with control analyses using random directions and permuted labels.
> ( 3 Methods / 3.5 Directional Ablation of Energy-Related Structure )
> AND  (  4 Result / 4.5 Directional Ablation )
> AND  ( Appendix F Additional Directional Ablation Analyses )
> - Expanded the TF–IDF lexical baseline and removed ambiguous terms such as "surface" and "n-gram". (in section 4 Result / 4.6 TF-IDF Baseline)
>
> **To address your questions:**
>
> > what applications could be supported?
>
> For example, localized regions could help study shifts toward higher-risk or coercive linguistic patterns without relying solely on keyword-based filtering. The observed geometric organization may motivate exploration of representation-space monitoring or steering approaches, in which generation trajectories are analyzed relative to geometric regions associated with different attributes. We emphasize that these are research directions rather than demonstrated applications.
>
> > in what sense could steering be called "non-manipulative"?
>
> We agree this wording was unclear. The term has been removed and the discussion (last paragraph in Section 6) has been revised for clarity.
>
> We sincerely appreciate the reviewer’s in-depth feedback and would be happy to clarify further if needed.

---

### Review · Reviewer_pmyr · 2026-02-15

**Summary Of Contributions:**

This paper investigates whether transformer sentence embeddings encode hierarchical structure aligned with human-defined cognitive/psychological states. The authors do this by constructing a dataset of 480 sentences with continuous scores between -5 and +5 (self-destructive states to integrative states) and categorical labels across seven categories (shadow, striving, conflict, activation, growth, clarity, and unity) and conducting linear and shallow non-linear probing of frozen transformer embeddings to decode the labels. In addition, the authors conduct permutation tests to determine whether the observed probe performance indicates a meaningful correspondence between the geometric structure of the embedding space and the annotated cognitive attributes. They also provide qualitative evidence through UMAP visualizations. They conclude that the geometric organization of model embedding space has a graded spectrum of human defined cognitive attributes.

**Audience:**

Yes

**Audience Explanation:**

Yes, this work would be interesting to people working at the intersection of cognitive science and machine learning. It may also be interesting to people working on interpretability, AI safety and alignment.

**Claims And Evidence:**

No

**Claims Explanation:**

## Issues related to data

### In section 2.1, the authors state -
```
we define a seven-tier taxonomy representing qualitatively distinct modes of consciousness-related cognitive experience.
```
The foundations of the definitions are a bit unclear, and could be better explained. How are these categories consciousness related? And how would this be different from existing work on decoding sentiment or emotions from transformers?


### In section 2.2 -
```
psychologically plausible phrasing
```
This is a bit unclear, and the authors could better reflect on what they mean here.
```
minimal reliance on explicit emotion words where possible.
```
Does this mean that there were some cases, where explicit emotion words were used? As the paper does not provide sentence content as mentioned in the "Data Availability" section, it is difficult to assess the extent of reliance on explicit emotion words. In addition, lack of sentence content availability also makes reproducibility of the results quiet challenging.

### In section 2.2 and 2.3 -
```
The annotations were performed manually by the author to ensure internal consistency across tiers.
```
```
Energy scores were assigned manually by the author to reflect the relative position of each sentence along the proposed low-to-high cognitive spectrum.
```
The 480 natural language sentences were constructed by the authors, the label categories were decided by the authors and labels were annotated by an author themselves. This brings up concerns about generalization as this study primarily demonstrates its claims about geometric organization of model embedding space with respect to the single annotator that annotated the data.

## Issues related to methods -

Although the TF-IDF baseline is a good addition, the evaluation could be more complete -
- The confusion matrix analysis is missing
- MSE for continuous regression is missing
- Accuracy is missing for classification
- The paper does not clearly state if the continuous regression result in this section is for shallow non linear probe or for linear probe.
- UMAP visualizations are missing

It is unclear why shallow non-linear regression for conducted only for continuous regression and not for classification.

Permutation test is conducted only for a single model which brings up concerns about generalizability.

## Issues related to references
I have mentioned the issues related to references in the "Requested Changes" section.

**Requested Changes:**

I would like to request the authors to -
- Clarify and quantify what "minimal reliance on explicit emotion words" means
- Complete the TF-IDF baseline evaluations as mentioned above.
- Provide permutation test results for all models
- Provide a clearer theoretical justification for why the seven-tier taxonomy is "consciousness-related" rather than being a sentiment hierarchy.
- Provide a justification for asymmetric experimental design (shallow non-linear probes only being used for continuous regression and not classification)


## Issues related to references-
Please correct me if I am wrong for the following mentions -
- The citation for "C-Pack: Packed Resources For General Chinese Embeddings" paper is wrong. The first author on that paper is Shitao Xiao.
- James Glass is not an author on the paper - "Probing Classifiers: Promises, Shortcomings, and Advances" - https://arxiv.org/abs/2102.12452
- I could not find the paper for the reference- "Taehee Kim et al. Interpretation of emotion representations in neural models. EMNLP, 2020."
- I am not sure if the paper - "Laura Weidinger et al. Ethical and social risks of harm from language models. ACL, 2021." was published at ACL 2021 - https://aclanthology.org/events/acl-2021/.
- I could not find the paper - "Niklas Muennighoff, Nils Reimers, Andreas Rücklé, and Iryna Gurevych. Sgpt: Gpt sentence embeddings for semantic search. In Proceedings of the 2022 Conference on Empirical Methods in Natural Language Processing (EMNLP), pp. 5881–5896. Association for Computational Linguistics, 2022." in the proceedings - https://aclanthology.org/events/emnlp-2022/ . And the paper on arxiv has a single author - https://arxiv.org/abs/2202.08904

---

> ### Author Response · Authors · 2026-02-26
> **Response to Reviewer pmyr**
>
> We thank the reviewer for the detailed feedback and for recognizing the potential relevance of this work to cognitive-science–oriented audiences. We have incorporated your suggestions into Revision 1.
>
> **Major changes include:**
>
> - In section 2.2 (Sentence Construction and Annotation), we removed ambiguous wording and clarified the intent and construction of the sentences, including how they differ from sentiment-focused datasets with concrete examples.
> - Provided a complete TF–IDF baseline evaluation, including linear and nonlinear probing, UMAP visualization, and confusion-matrix analysis. ( in Section 4.6 TF-IDF Baseline) AND (Appendix G Additional Analyses for the TF–IDF Baseline)
> - Provide permutation test results for all models. (in section 4.4 Statistical Significance via Permutation Tests) AND (E Additional Permutation Test Results)
> - Removed the “consciousness-related” wording and replaced it with more neutral cognitive terminology to avoid ambiguity and confusion.
> - Added shallow nonlinear classification probes (MLP) across all models for methodological consistency with regression experiments. ( in Section 4.2 Decodability of Cognitive Tiers)
> - Corrected several citation issues and updated references.
>
> In addition, we clarified that the annotations represent an internally consistent exploratory taxonomy constructed by a single annotator, and do not aim to claim universal psychological structure.
>
> **To address your concern:**
> > lack of sentence content availability also makes reproducibility of the results quiet challenging.
>
> We included the raw sentence data in the Revision 1 supplementary package.
>
> We sincerely appreciate the reviewer’s careful reading and detailed suggestions. Please let us know if further clarification would be helpful.

---

### Decision · Action_Editor_s7Ey · 2026-04-02

**Recommendation:** Reject

**Additional Comments:**

If the authors wish to resubmit this work, I suggest that they conduct a more thorough study involving multiple hypothesis-blind raters for the concepts in question, and evaluating features like interrater reliability.

**Audience:**

Yes

**Audience Explanation:**

The reviewers generally agreed that some in the TMLR audience might be interested in these findings from practical perspectives or for connections to cognitive science.

**Claims And Evidence:**

No

**Claims Explanation:**

While the reviewers generally agreed that the claims of the paper have been improved by the revision (e.g., reducing the consciousness language), there continued to be some concern about definitions of the concepts used. This includes both the distinction from extant concepts like sentiment that appear to overlap with the dimension identified, and more importantly the fact that the core findings of the paper rely entirely on a single annotator, who is also an author of the paper. I concur with this concern. It is best practice when annotating data to have multiple hypothesis-blind annotators (ideally labeling using established criteria or definitions from the literature), and assess their inter-rater reliability. Otherwise, it is difficult to be confident that the annotations truly reflect the putative cognitive constructs they are intended to. This type of more rigorous analysis seems necessary to fully justify the claims.

Although it was not the primary deciding factor, there is one other cause for concern. Although the authors edited the paper to fix the reference issues raised by reviewer pmyr, they did not provide a clear explanation for how they had many incorrect citations initially, including references that did not appear to exist at all and have since been removed. This raises concerns about the overall research practices underlying this work.

**Resubmission Of Major Revision:**

The authors may consider submitting a major revision at a later time.